# Probabilistic Safety Analysis of the Liquefaction Hazard for a Nuclear Power Plant

**Tamás János Katona** [1,*] and **Zoltán Karsa** [2]

1   Faculty of Engineering and Information Technology, University of Pécs, 7624 Pécs, Hungary
2   NUBIKI Nuclear Safety Research Institute, Ltd., 1121 Budapest, Hungary; karsa@nubiki.hu
*   Correspondence: katona.tamas@mik.pte.hu

**Abstract:** Liquefaction hazard safety is essential for operating nuclear power plants where the elimination of hazards via engineering measures is not practicable. For this, the core damage frequency should be evaluated via integration of the liquefaction hazard into the seismic probabilistic safety analysis. In the seismic probabilistic safety analysis, the maximum horizontal acceleration is used as the intensity measure and as the engineering demand parameter for a simple calculation of failure rates. According to the studies performed for the Paks Nuclear Power Plant, loss of emergency service water supply due to relative settlement of adjacent structures and structural and functional failures due to tilting are the dominating failure modes. To integrate these failure modes into a seismic probabilistic safety analysis, hazard and fragility should be evaluated as functions of properly identified intensity measures and engineering demand parameters, preferable the maximum horizontal acceleration. Since a generic procedure does not exist in nuclear practice, based on the analyses for the Paks Nuclear Power Plant, two practical options are proposed for integration of the liquefaction hazard into a seismic probabilistic safety analysis, and for the calculation of annual probability of failure of critical structures.

**Keywords:** liquefaction; nuclear safety; probabilistic safety analysis; fragility; settlement

## 1. Introduction

Soil liquefaction is a possible secondary phenomenon of earthquakes at soil sites of nuclear power plants (NPPs) which can affect the safety of the nuclear powerplant. In the case of site selection for a new NPP, one of the aspects of site investigation is whether the site soil layers are susceptible to liquefaction [1] and engineering measures should be implemented to avoid this hazard. For screening out the hazard, at the beyond-design basis earthquake level, a conservatively calculated value of the factor of safety to the liquefaction ($FS_L$) can be applied [2].

For NPPs operating at soil sites, the liquefaction hazard and related safety issues should be investigated, and plant safety should be assured since the elimination of the hazard via engineering measures could be not practicable. In the case of the Paks NPP in Hungary, extensive investigations were performed during post-Fukushima stress tests [3]. Re-evaluation of the liquefaction hazard has also been mentioned in the post-Fukushima action plan of the Netherlands [4].

Considering the practice of operating NPPs, significant efforts have been made to evaluate the liquefaction hazard and plant safety at the Paks NPP, Hungary, because of obvious liquefaction susceptibility of the Holocene sediments at the site. These investigations cover detailed field and laboratory tests, and evaluation of the liquefaction hazard using well-known deterministic and probabilistic methods based on SPT and CPT tests [5–8], and the nuclear industry guidance developed by EPRI [9]. The structural integrity of safety-classified building structures has been proven for the liquefaction effects caused by a beyond-design basis earthquake with 0.25 g maximum horizontal acceleration at free

field (PGA) assuming gross liquefaction [8,10,11]. A seismic probabilistic safety analysis has also been performed, where the liquefaction hazard has also been considered [12]. The presentation and discussion of the literature used on liquefaction hazard assessments and analyses of structural response to liquefaction are beyond the scope of this study. This is published in [5–8,10,11]. Examples of relevant literature sources used in investigations of the Paks NPP are [13–24].

Because of the variety of safety-related building structures, a plant's responses to liquefaction effects are rather complex. At the Paks NPP, there are various shallow-founded buildings with different sizes, foundation shapes, embedment depths, and masses. There are water intake structures, diesel-generator buildings, auxiliary buildings, electrical sub-station structures, etc., and there are the buildings with very large bearing pressures, such as the containment. The plant structures are connected by communication lines. These are surface and buried pipes and cables, some of which are placed in underground cable and piping tunnels. The differential movement of the adjacent structures is a significant failure mode that can damage safety-related communications. The response of the structures transfers the effects to the systems and their components with very complex nuclear technology. Tilting of the reactor axis due to tilting of the reactor building can affect reliable insertion of the control rods into the reactor core.

To safely shut down a nuclear reactor, heat removal from the reactor core and the retaining function of the containment should be ensured. The aim of the performed analyses was to determine if these fundamental safety functions would fail due to liquefaction.

Deterministic analyses that have been performed for the plant buildings for gross liquefaction have shown that the dominating liquefaction effect was the tilting of the buildings due to differential settlement on non-homogenous soil strata. Structural evaluations have shown that the tilting effect on the structural integrity was below the acceptable level of 0.003 allowed by the relevant design code [25]. Thus, the structural failures that would cause loss of containment function could be deterministically screened out. Similarly, the tilting of the reactor axis that would hinder the function of control rods could also be screened out. Considering emergency heat removal from the reactor core, where the emergency feedwater pipe enters into the main building is a critical location (see, Figure 1). If the gap between the wall and the pipe is closing due to a difference between the settlement of the building and the pipe, the pipe can break with consequent loss of emergency heat removal. This failure could not be a priori excluded for beyond-design basis earthquakes. The pipe outside of the building is in dry sand above the ground water table. An analysis showed that the pipe settlement outside of the building is approximately equal to the settlement of the free field due to liquefaction.

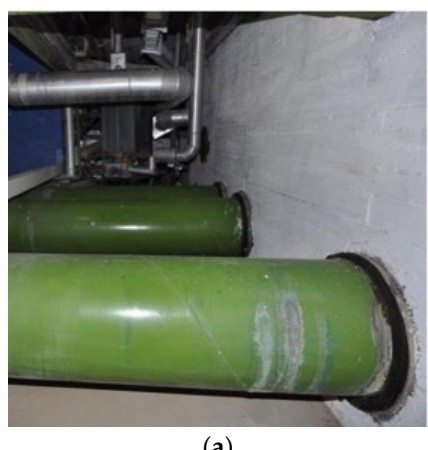

(a)

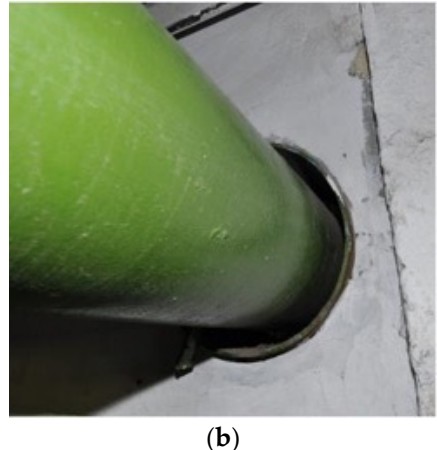

(b)

**Figure 1.** (**a**) Intersection of the pipelines of the emergency service water system with the wall of main building; (**b**) gap between the wall and pipe.

Although the conclusions on the safety of the deterministic analyses for beyond-design basis earthquakes are generally positive, the nuclear regulation (see example in [26]) requires justification of plant safety in terms of core damage frequency due to liquefaction that could be defined by a seismic probabilistic safety analysis. The analysis should be extended to annual probabilities $10^{-7}$/year to demonstrate that a beyond-design basis earthquake would not cause a cliff-edge effect.

In the established seismic probabilistic safety analysis (SPSA) methodologies (see [27,28]), the PGA is used for intensity measure of the seismic hazard as a single scalar variable. The PGA is also used as the independent variable in the fragility function. According to [27,28], the annual probability of failure $\lambda_{ij}$ of the SSCs can be calculated via generic equation:

$$\lambda_{ij} = \int_0^\infty \frac{dH_i(a)}{da} F_j(a) da, \tag{1}$$

where $dH_i(a)/da$ is the $i$th representation of the annual probability density of exceedance for maximum horizontal acceleration, $a$ (PGA); $dH_i(a)da = d\lambda(a)$ is the annual frequency for maximum horizontal acceleration to be within the interval $[a, a + da]$; and $F_j(a)$ is the $j^{th}$ fragility curve versus maximum horizontal acceleration at the free field, $a$. The damage state frequency, $\lambda$, is the weighted average value by the probabilities associated with fragility and hazard curves. Here, $H_i(a)$ is defined by the probabilistic seismic hazard analysis. The $F_j(a)$ can be defined analytically or by tests. The plant as a complex system is modeled by event trees and fault trees, and the annual probability of unaccepted plant performance is calculated using Boolean model of the entire plant [27,28].

Integration of the liquefaction hazard into the seismic PSA requires an appropriate definition of the hazard and fragility for liquefaction effects. For full compliance with the SPSA methodology, both the liquefaction hazard curve and the fragility curve should preferably be functions of the PGA as a single variable.

Regarding the liquefaction hazard curve, it would be reasonable to consider adaptation of the probabilistic liquefaction hazard analysis (PLHA) procedure developed by Kramer and Mayfield (2007) [16] in the framework of performance-based earthquake engineering methodology [29]. Despite the practical applications of this approach, it has not been applied for nuclear facilities since the calculation of the core damage frequency requires complex modeling of the response of the plant, as done in an SPSA. Nevertheless, since the methodology of Kramer and Mayfield [16] generates the annual frequency of exceedance $\Lambda_{FS_L^*}$ of $FS_L \leq FS_L^*$, this hazard curve should be used for the calculation of the failure rate of SSCs. For this, the fragility should be expressed as a function of engineering demand parameter that should be correlated to the $FS_L$. Moreover, for the integration into the SPSA, the liquefaction hazard curve should be linked to the seismic hazard curve.

Calculation of the failure rates within the framework of an SPSA via Equation (1) is also possible if the engineering demand parameter due to liquefaction for the critical SSCs could be approximately expressed as a function of the PGA. In this case, the failure rates of the SSCs could be calculated for the seismic hazard curve as usual on the basis of the SPSA.

In the earlier SPSA made for the Paks NPP, the fragility for liquefaction was considered in a rather simplified manner, assuming loss of function of the affected SSCs at a certain value of $FS_L$. This simplification of the fragility function was motivated by a lack of adaptable industry practice. According to the EPRI guidance [30], the conditional probability of failure of buried pipeline was calculated and presented in form of the conventional double lognormal function versus PGA. For the generation of the fragility curve, 500 simulations were needed. Unfortunately, the fragility curves provided by the EPRI could not be used for the identified Paks NPP emergency feedwater piping, since the demand on the pipe was caused by the interaction between the main building and buried pipeline. Analytical development of the fragility curve for the coupled soil-structure model and for the structure–structure interaction would require enormous computational effort.

Two options could be identified for the integration of liquefaction hazard into an SPSA:

1. To use the liquefaction hazard curve versus $FS_L$ obtained by the procedure of Kramer and Mayfield [16], express the fragility as a function of properly selected engineering demand parameter that can be correlated to $FS_L$, and link the liquefaction hazard curve to the seismic hazard curve for the consequent integration of the failure rates due to liquefaction into the evaluation of the core damage frequency via SPSA.

2. Find a simplified, approximate liquefaction hazard as well as the fragility of SSCs as functions of the PGA that are based on the established publications on the liquefaction phenomenon. Calculate the failure rates of SSCs using these hazard and fragility estimates in the framework of a seismic PSA.

The objective of this study is to construct a practical procedure, considering the options outlined above, for integrating the liquefaction hazard and its consequences into a seismic probabilistic safety analysis. The novelty of the research is the procedure itself, which is proposed using the known methodology of seismic PSA, known and published results of research on the liquefaction phenomenon, and studies performed for the Paks NPP mainly by contribution of the authors. Finally, the applicability of the procedure is demonstrated through a focused analysis of liquefaction-induced failure modes at the Paks NPP, Hungary.

## 2. Basis of the Method

Site information relevant to the methodological considerations and basic results of liquefaction studies performed for the Paks NPP by one of the authors are briefly presented below.

### 2.1. Seismic Hazard

The Paks NPP site is in a low-to-medium seismicity region, in the middle of the Pannonia Basin, in the Danube Floodplain (46.34 N, 18.51 E). A probabilistic seismic hazard analysis has been performed for evaluation of the site seismic hazard curves (Figure 2) [31]. The seismic hazard curve has been defined based on the modeling of seismogenic sources and utilizing the R-CRISIS PSHA methodology, see, [32].

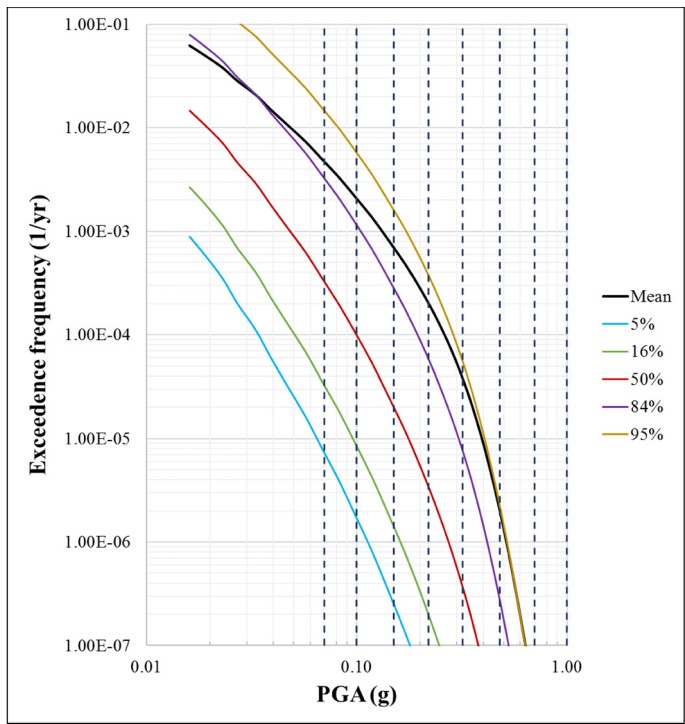

**Figure 2.** Seismic hazard curve.

In the seismic PSA methodology, the evaluation of unacceptable performance is calculated by dividing the seismic demand in the hazard curve into 6 to 8 intervals and the midpoint values of the peak ground accelerations and annual rates in each interval are assumed to represent the demand in the entire interval. The division of the seismic demand into intervals is shown in Table 1 and Figure 2.

**Table 1.** Division of the seismic demand into calculation intervals.

| Initiating Event | Intervals of PGA (g) | Midpoint Rates (1/Year) |
| --- | --- | --- |
| SEIS1 | 0.07–0.10 | $2.66 \times 10^{-3}$ |
| SEIS2 | 0.10–0.15 | $1.37 \times 10^{-3}$ |
| SEIS3 | 0.15–0.22 | $4.96 \times 10^{-4}$ |
| SEIS4 | 0.22–0.32 | $1.62 \times 10^{-4}$ |
| SEIS5 | 0.32–0.48 | $3.45 \times 10^{-5}$ |
| SEIS6 | 0.48–0.70 | $1.91 \times 10^{-6}$ |

The failure rate of SSCs can be calculated via discretization of the Equation (1).

A detailed presentation and discussion of the SPSA procedure is beyond the scope of this study; however, it is described in the references, for example, [27,28].

### 2.2. Liquefaction Hazard Studies

The liquefaction hazard was previously investigated in the early nineties and re-evaluated several times in the framework of periodic safety assessments and post-Fukushima stress tests. A comprehensive geotechnical survey has been made at the site. There are nearly 500 boreholes and other test points, and more than 100 groundwater-monitoring wells. A site geotechnical survey includes mapping the soil stratigraphy, in situ definition of soil properties, full scope laboratory testing of samples, cyclic triaxial and resonant column test, SPT, CPT, CPTu, and SCPT. The geotechnical conditions at the site are illustrated in Figure 3, which show the soil description and parameters for the test location at the site [5].

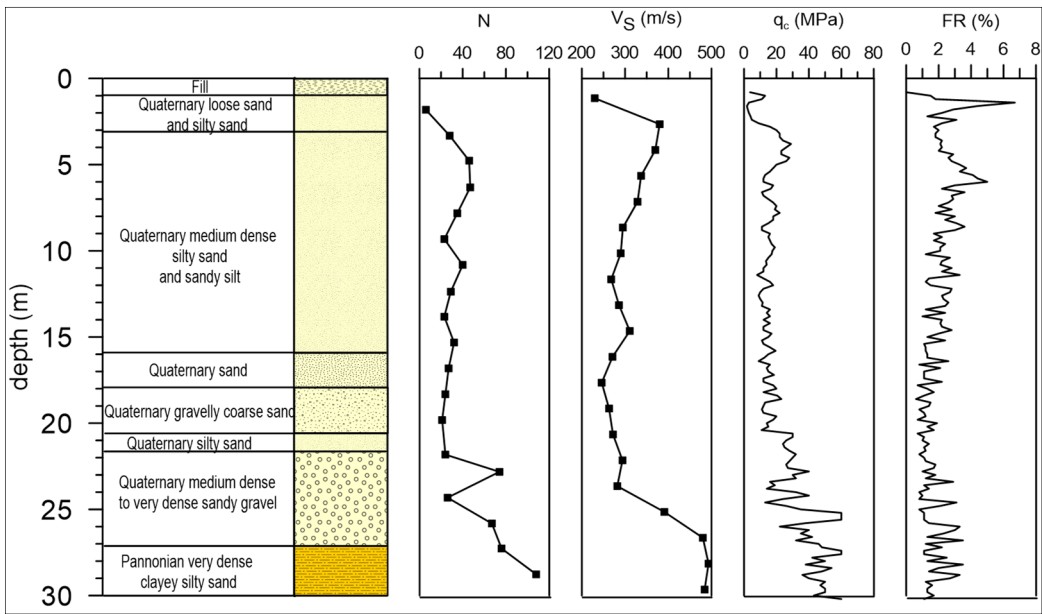

**Figure 3.** Example for soil properties at the site of the Paks NPP (N is the SPT count, vs. is the shear wave velocity, q is the cone resistance, and FR is the friction ratio). The water table is at −8 m.

Altogether, 17 liquefaction hazard evaluation methods have been applied for the site including the full probabilistic method of Kramer and Mayfield [16] shown in Figure 4 [31]. The mean annual rate $\Lambda_{FS_i^*}$ of non-exceedance of $FS_L < FS_L^*$ is calculated for all possible

$a_{max}$, $M_w$ pairs and combined with the Boulanger and Idriss CPT-based method [13] for $FS_L$ by the following equation:

$$\Lambda_{FS_L} = \sum_{j=1}^{j=N_{M_w}} \sum_{i=1}^{i=N_{a_{max}}} P[FS_L < FS_L^*|(a_{max}, M_{w,i})] \Delta\lambda_{a_{max,i} M_{w,j}} \qquad (2)$$

where $N_{M_w}$ and $N_{a_{max}}$ are the numbers of magnitude and peak ground acceleration increments, respectively; $P[FS_L < FS_L^*|(a_{max}, M_{w,i})]$ is the conditional probability for $FS_L < FS_L^*$; $\Delta\lambda_{a_{max,i} M_{w,j}}$ is the *i*th incremental mean annual rate of exceedance for intensity measure $a_{max}$, $M_w$.

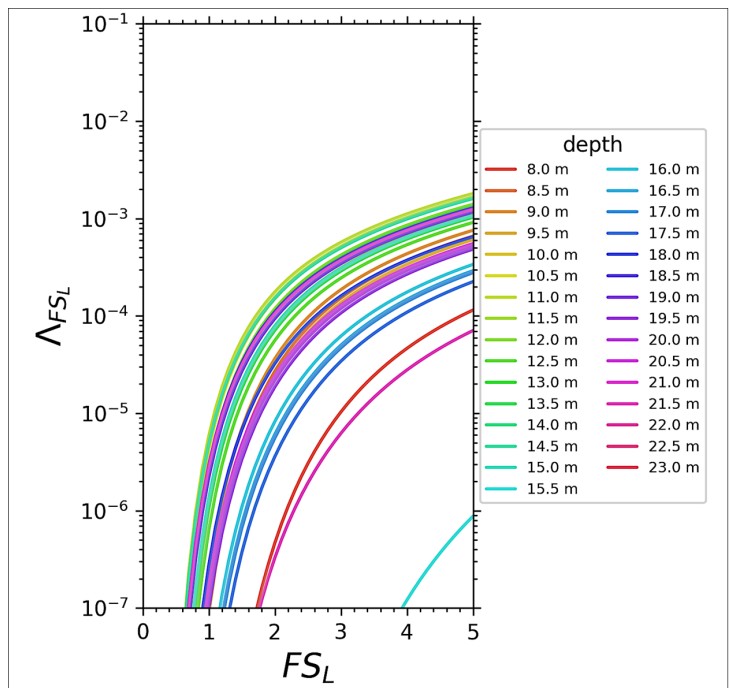

**Figure 4.** Mean annual rate of non-exceedance for $FS_L$ for the NPP Paks site, calculated applying the method of Kramer and Mayfield [16] combined with the Boulanger and Idriss CPT-based method [13].

## 2.3. Settlement Analysis

In the case of the Paks NPP, the settlement of the free field as well as the settlement of the structures have been analyzed assuming different models for liquefaction triggering and settlement evaluation. For example, for the free field settlement evaluation, the methods given in Table 2 have been applied.

**Table 2.** Methods applied for the evaluation of the settlement at the free field at the study site.

| Method for Liquefaction Evaluation | Method for Settlement Evaluation |
|---|---|
| Youd et al. (2001), Idriss and Boulanger (2008) | Tokimatsu and Seed (1987)—SPT |
| Youd et al. (2001), Idriss and Boulanger (2008) | Ishihara and Yoshimine (1992)—SPT [17] |
| Cetin et al. (2004) | Wu and Seed (2004)—SPT |
| Robertson and Wride (1998), Moss et al. (2006), Idriss and Boulanger (2008) | Zhang et al. (2002)—CPT [18] |
| Cetin et al. (2009)—SPT | Cetin et al. (2009)—SPT |

References indicated in Table 2 are for information only. Those are included into the reference list that are directly used in the calculations.

## 3. The Method

### 3.1. Procedure Based on the PLHA Hazard Curve

According to the experiences presented in the introduction, the non-uniform settlement of the building determines the demand and the fragilities. The settlement can be calculated based on the post-liquefaction volumetric strain $\varepsilon_v$. Ishihara and Yoshimine [17] developed a set of empirical curves $\varepsilon_v(FS_L)$. The $\varepsilon_v(FS_L)$ is parametrized by relative density of the soil layer, $Dr$. The relation $\varepsilon_v(FS_L)$ has also been given in Zhang et al. [18].

First, select a layer $i$ with $Dr_i$, and fix the $j^{th}$ hazard level $\lambda_j$ at the $j^{th}$ midpoint rate of seismic hazard (Table 1). For this layer and hazard level, the $FS_{Li,j}$ is defined by the liquefaction hazard curves $\Lambda_{FS_L,i}$, see Figure 4. Using this $FS_{Li,j}$, the volumetric strain $\varepsilon_{v,i}(FS_L)$ can be calculated. This calculation can be performed for all hazard levels equal to the midpoint rate in the $j^{th}$ demand interval $\lambda = \lambda_j$, (see Figure 2 and Table 1). The settlement for the $i$th layer at $\lambda = \lambda_j$ will be:

$$S_{j,i} = \varepsilon_{vi,j}\left(FS_{Li,j}\right)\cdot\Delta z_i\cdot P_{Li,j} \tag{3}$$

where $\varepsilon_{vi,j}$ is the volumetric strain of the $i$th layer and $\Delta z_i$ is the thickness of the $i$th layer. The multiplier $P_{Li,j} \approx 1/\left[1 + \left(FS_{Li,j}/0.9\right)^6\right]$ in Equation (3) accounts for the probability of the liquefaction occurrence in the $i$th layer [18]. For the sake of simplicity of writing, the relative density $Dr$ that is also a parameter of the function $\varepsilon_{vi,j}$ is omitted.

Thus, by combining the hazard curve $\Lambda_{FS_L,i}$ with the function $\varepsilon_{vi,j}\left(FS_{Li,j}\right)$, the mean rate of non-exceedance for $\varepsilon_{v,i}$ can be generated, as shown in Figure 5 for two layers with $Dr$ equal to 70% and 80%, respectively.

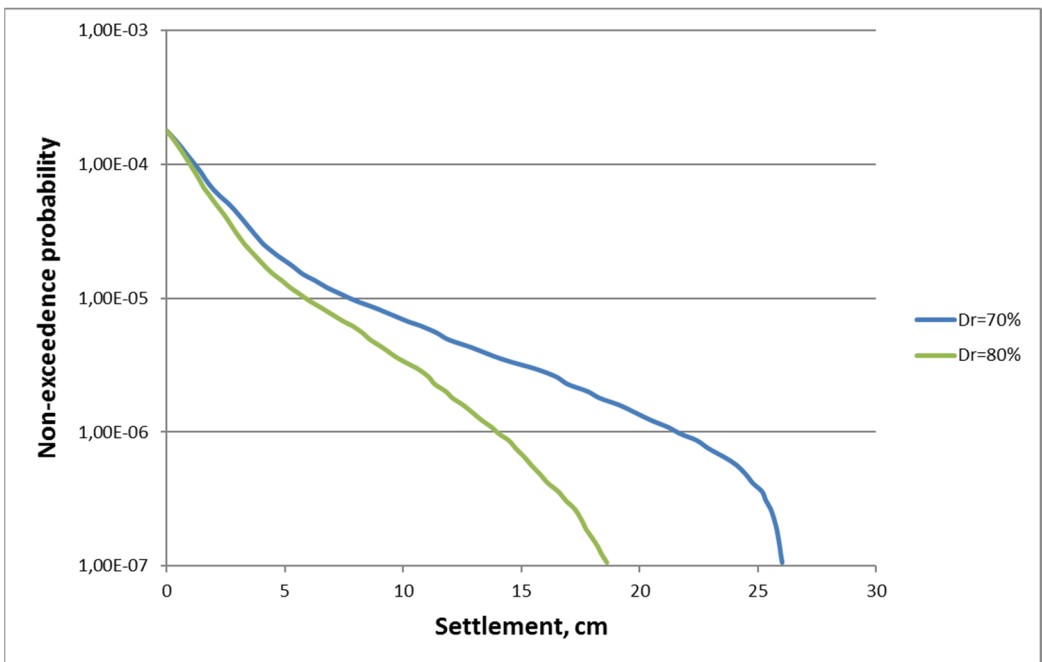

**Figure 5.** Mean annual rate of non-exceedance for $\varepsilon_{v,i}$ at the hazard level $\lambda^*$, calculated via Equation (5) using the mean annual rate of non-exceedance for $FS_L$ shown in Figure 4 for the NPP Paks site.

Summarizing for all layers, the total settlement for the $j^{th}$ hazard level $\lambda = \lambda_j$ will be:

$$S_j = \sum_{i=1}^{i=N} S_{j,i} = \sum_{i=1}^{i=N} \varepsilon_{vi,j}\cdot\Delta z_i\cdot P_{Li} \tag{4}$$

where $N$ is the total number of layers.

Instead of the relation of Ishihara and Yoshimine [17] and Zhang et al. [18], in the above procedure, the empirical function for the calculation of the volumetric strain, $\varepsilon_v$, proposed by Juang et al. (2013) [19] can be used. Here, the $\varepsilon_v$ is expressed as a function of $FS = FS_L$. The soil layer is identified by the corrected cone tip resistance, $q$, instead of the relative density $Dr$.

The equation for $\varepsilon_v$ is as follows:

$$\varepsilon_v(\%) \begin{cases} 0 & if & FS \geq 2 \\ \min\left(\frac{a_0 + a_1 ln(q)}{\frac{1}{(2-FS)} - (a_2 + a_3 ln(q))}\right), \; b_0 + b_1 ln(q) + b_2 ln(q)^2 & if & 2 - \frac{1}{a_2 + a_3 ln(q)} < FS < 2 \\ b_0 + b_1 ln(q) + b_2 ln(q)^2 & if & FS \leq 2 - \frac{1}{a_2 + a_3 ln(q)} \end{cases} \quad (5)$$

where $FS = FS_L$; $\varepsilon_v$ is the volumetric strain; $q$ is the corrected cone tip resistance; $a_0$, $a_1$, $a_2$, and $a_3$, and $b_1$, $b_2$, and $b_3$ are constant parameters as per [19].

Using Equations (4) and (5), the total settlement for the $j^{th}$ hazard level, $\lambda = \lambda_j$, can be calculated as:

$$S_j = \sum_{i=1}^{i=N} S_{i,j} = \sum_{i=1}^{i=N} \varepsilon_{vi,j}(q_i, \; FS_{Li,j}) \cdot \Delta z_i \cdot P_{Li,j} \quad (6)$$

The factor of safety to liquefaction $FS_{Li,j}$ for the fixed hazard level can be read from Figure 4.

The annual rate of failure of plant SSCs due to the settlement for this hazard interval $j$ can be calculated as:

$$\lambda_{fail, \; j} = P\left(S_j \geq S_{fail}\right) \cdot \lambda_j \quad (7)$$

Here, the $P(S) = P\left(S_j \geq S_{fail}\right)$ is the conditional probability of failure versus settlement, $Sj$ in the $j^{th}$ seismic demand interval. This is practically the expression of the discretized form of Equation (1) for the $j^{th}$ interval.

Summarizing the above considerations for the calculation of liquefaction-induced failures and integration of these failures into the framework of a seismic probabilistic safety analysis, the following calculation procedure should be applied:

1. Select the hazard level to the midpoint annual rate of the $j^{th}$ demand interval according to Table 1.
2. For this fixed annual rate, read the $FS_{Li,j}$ for the layer $i$ from the PLHA hazard curve.
3. Calculate the volumetric strain and the settlement for the layer $i$, with CPT tip resistance, $q_i$, using Equations (4) and (5).
4. Calculate the total settlement for the soil column using Equation (6).
5. Use the settlement evaluated for the selected midpoint annual rate in the calculation of the probability of failure via Equation (7). The failure condition should be a function of the settlement. (The simplest assumption is to select the standard allowable for the tilt due to settlement, or the limiting value for the differential settlement allowed by engineering consideration, as it will be shown in Section 5).

Consequently, the failure rates of critical SSCs due to liquefaction can be accounted for in the calculation of the core damage frequency via seismic PSA, since the liquefaction-induced failures are calculated for midpoint rates of the intervals of the seismic hazard.

### 3.2. Procedure for Integration Based on the Seismic Hazard Curve and Peak Ground Acceleration

Accepting some compromises, the above procedure would allow integration of the liquefaction hazard into the seismic PSA based strictly on the peak ground acceleration formalism as in Equation (1). The procedure is as follows:

1. Use the seismic hazard curve (Figure 2) that defines for any hazard level $\lambda$ the maximum horizontal acceleration $a_{max}(\lambda)$.

2. Select the hazard level to the midpoint annual rate of the $j^{th}$ demand interval, according to Table 1.

3. Calculate the factor of safety to liquefaction for each soil layer and for the selected hazard level, $FS_{Li,j}$. For this, calculate the $FS_L$ for the peak ground acceleration value corresponding to the selected midpoint rate of seismic demand (Table 1) by the method of Robertson and Wride [14]. The selection of the method [14] is justified by the experience of analyses performed for the Paks NPP.

Our goal is to express the $FS_{Li,j}$ as a function of the $a_{max}$. Here, certain assumptions should be made, since the $FS_L$ also depends on the magnitude scaling factor (MSF), $FS_L(a_{max}, MSF(M_w))$. Disaggregation of the seismic hazard provides the weights of contribution for different magnitudes to any hazard level $\lambda$, see Figure 6. A characteristic magnitude value could be selected based on the weight distribution. For any hazard level, the mean magnitude could be a conservative selection, as it has been done, for example, by Katona et al. in [7]. Thus, the dependence of the $FS_L$ on the MSF($M_w$) could be eliminated via selection of a single value of the $M_w$ for the fixed hazard level.

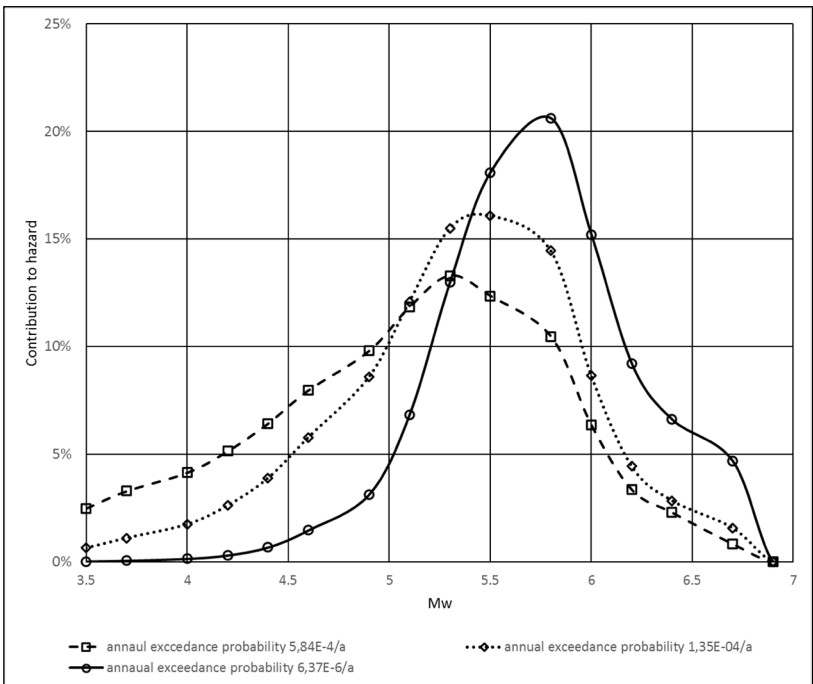

**Figure 6.** Contributions of different magnitudes to the seismic hazard.

4. Calculate the volumetric stain $\varepsilon_{vi,j}(q_i, FS_{Li,j})$ and the settlement, $S_j$, for $j^{th}$ hazard level $\lambda = \lambda_j$ using Equations (5) and (6). Considering the above simplification, the volumetric strain will be a function of the $FS_{Li,j}$ that is a function of the $a_{max}(\lambda)$, only.

5. Calculate the total settlement that is also a function of the $a_{max}(\lambda)$, only.

6. Define the fragility in terms of settlement. (Examples are given in Section 5). Due to the above simplifications, the fragility is also function of $a_{max}(\lambda)$.

7. Calculate the failure rates of critical SSCs due to liquefaction for the $j^{th}$ hazard level via Equation (7). This will exactly correspond to the seismic hazard curve. Thus, the full formal compliance with the methodology of seismic PSA is achieved.

## 4. Evaluation of the Uncertainties

There are different sources of uncertainty in the above calculations for the annual rate of unaccepted performance of the plant.

In the procedures outlined in Section 3.1, the mean liquefaction hazard curve, which is derived based on the PLHA analysis, is used to account for the aleatory and epistemic

uncertainties of the definition of the hazard. Therefore, additional considerations should be made to evaluate the uncertainties of settlement and settlement-induced failures. According to Juang et al. [19], the uncertainty of Equation (5) can be evaluated via introducing the model bias factor $M$ that corrects the settlement prediction $S_c = M \cdot S_P$. The bias factor can be calibrated using field observations. The database of field observations and the maximum likelihood analysis are presented in the [19].

In the study case of the Paks NPP, a specific method is applied for the calibration that is based on the large number of performed deterministic analyses briefly presented in Section 2. Consequently, it is sufficient to assess the uncertainty of the evaluation for settlement as proposed by Juang et al. [19], and the uncertainty of the fragility function.

In the procedures outlined in Section 3.2, the probabilistic seismic hazard analysis provides the seismic hazard curves for different confidence levels, see Figure 2. This accounts for the aleatory and epistemic uncertainties of the definition of the seismic hazard. The uncertainty for the settlement can be assessed as proposed by Juang et al. [19].

In both procedures (Sections 3.1 and 3.2), the uncertainty of the fragility function is an open issue. If the condition of the failure is defined based on the stresses, strains, etc. allowable by a design standard, for example, by [25], this uncertainty is negligible as compared with other contributors to uncertainty. The uncertainty of engineering limitations used as failure criterion could also been neglected, see examples in Section 5.

## 5. Results of the Case Study

Below, for the case of the NPP Paks, Hungary, the PGA-based procedure for the integration of liquefaction hazard into a seismic PSA are applied, as presented in Section 3.2.

### 5.1. Calibration of the Predicted Settlements

Regarding the calibration method, the concept of Juang et al. [19] was adopted. The set of settlement results calculated in earlier studies for the free field and for the plant structures were used for the calibration of the settlement prediction. The results of the calibration of predicted settlements to the PGA values are shown in Table 3 for two characteristic points at the plant, A4 and 3e. The soil-structure calculation performed for the main reactor building of the Paks NPP for a beyond-design basis earthquake resulted in a 0.21 m differential settlement obtained for the opposite edges of the foundation [10]. The calculation of the settlement according to the procedure in Section 3 would predict this settlement value for peak ground acceleration (0.445 g) that can be judged as realistic. Strictly speaking, Equation (5) used in the above procedures is valid for the settlement of the unloaded soil. As an approximation, the behavior of the main reactor building can be evaluated by the method developed above, as a result of the calibration of the prediction based on the results of a comprehensive soil-structure analysis for the building settlement.

**Table 3.** Results of calibration.

| Target Settlement | | Peak Ground Acceleration Level (g) |
|---|---|---|
| Location | Settlement (cm) | |
| A4, loaded by the building | 21 | 0.445 |
| 3e, unloaded soil | 0.6 | 0.381 |
| 3e, loaded by the building | 4.4 | 0.503 |

### 5.2. Tilting of the Main Reactor Building

A detailed finite element analysis of the main reactor building has been performed for the demand due to tilting caused by differential settlement. This analysis justified acceptable performance from the point of view of structural integrity and leak tightness of the containment [8,10,11]. The tilting of the reactor vertical remains below the allowable limit. As per design code [25], the allowable tilt (0.003) can be used as the failure criterion for the main reactor building. This is a conservative failure criterion. For the reactor

pressure vessel vertical axis, a tilt of 0.0014 is allowed. These allowable limits have been accepted as failure criteria in the fragility estimation.

The tilt of the main rector building versus peak ground acceleration calculated via the procedure presented in Section 3.2 is shown in Figure 7. The uncertainty of the tilting was evaluated according to Section 4.

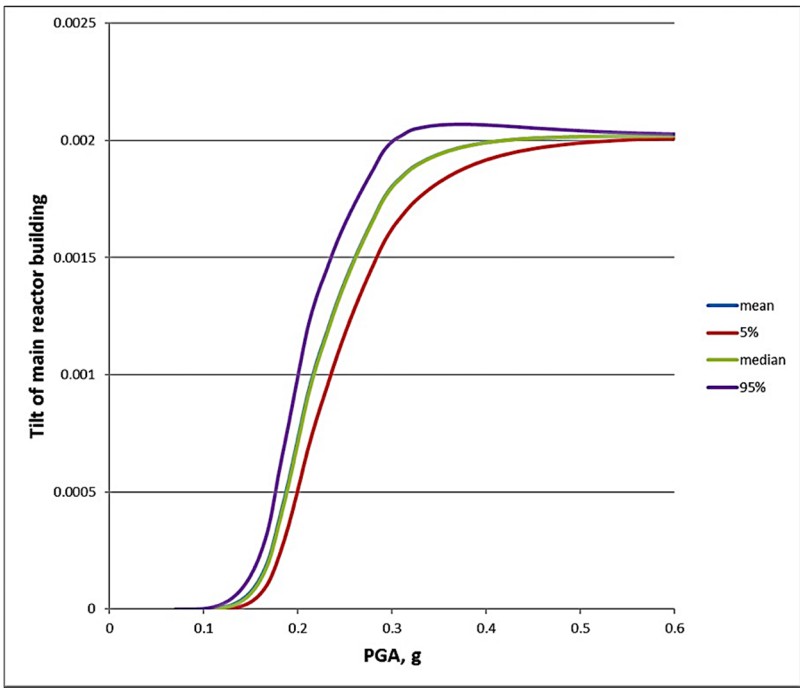

**Figure 7.** Tilt of the reactor building due to differential settlement versus peak ground acceleration.

According to this result, the failure criterion of the main reactor building cannot be reached despite the increasing peak acceleration. This complies with the EPRI [30] conclusion that the asymptotic failure probability may not reach the value of one, despite increasing peak ground acceleration, since the volumetric strains, and consequently, the settlement remains limited despite increasing peak ground acceleration. For volumetric strain, the maximum packing is the obvious theoretical limit. Due to these physical constraints, the settlement-related fragility curves could not be properly conveyed by equivalent parameters of a double lognormal probability distribution.

### 5.3. Failure of the Emergency Service Water Piping due to Differential Settlement

The critical section of the emergency service water piping is shown in Figure 1. For the sake of simplicity, closing the 0.05 m wide gap between the piping and building wall is selected as the failure criterion, although the pipe is ductile and can sustain relatively large deformation.

The settlement of the main building and unloaded free field settlements versus peak ground acceleration at the critical section of the emergency service water piping is shown in Figure 8.

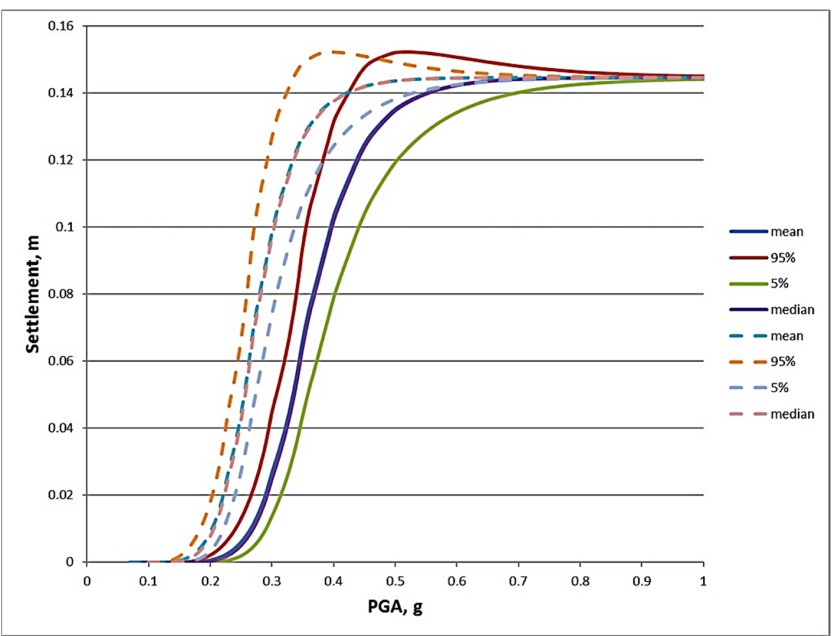

**Figure 8.** Loaded by the main building (dashed line) and unloaded, free field (solid line) settlements versus peak ground acceleration at the critical section of the emergency service water piping.

Applying the idea proposed by Juang et al. [19], distribution function for the gap between the pipe and the wall can be written as:

$$P[s_2 > s_1 + 5 \text{ cm}] = \Phi\left\{\frac{ln(s_1) - \nu_1}{\xi_1}\right\} \cdot \left[1 - \Phi\left\{\frac{ln(s_1 + 5 \text{ cm}) - \nu_2}{\xi_2}\right\}\right] \qquad (8)$$

where $s_1$ is the unloaded or free field settlement; $s_2$ is the settlement if the soil column loaded by the building; $\nu_1$, $\nu_2$ and $\xi_1$, $\xi_2$ are the logarithmic mean and variance of the predicted settlements, respectively. This exceedance probability distribution function of the gap versus settlement at the free field is shown in Figure 9.

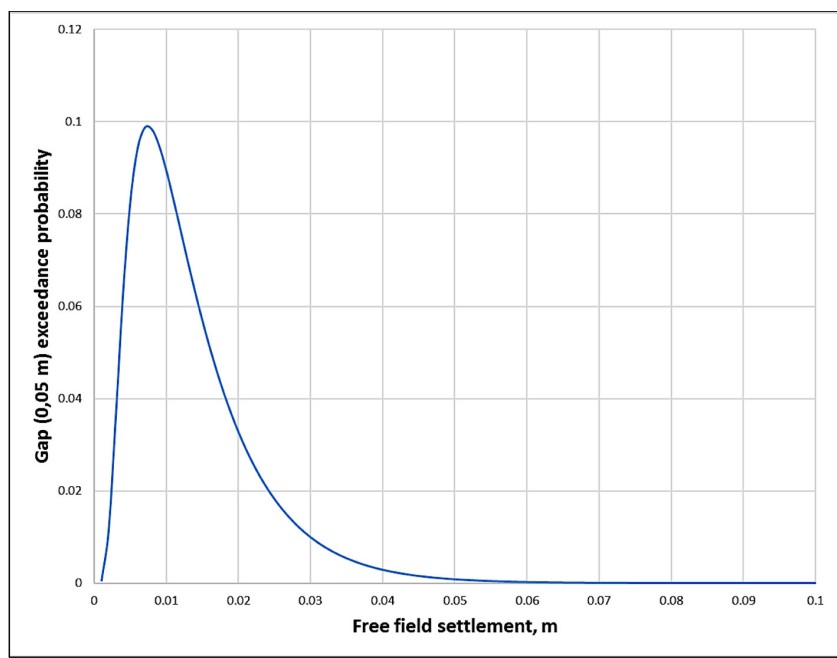

**Figure 9.** Exceedance probability distribution function of the gap versus settlement at the free field.

The total probability of exceedance of the gap size 0.05 m can be obtained by numerical integration of the curve in Figure 9. By performing the calculations for all peak ground acceleration intervals (see Table 1), the exceedance probability for the gap 0.05 m can be calculated. The exceedance probability for closing the gap between the emergency service water pipe and the building wall is plotted in Figure 10.

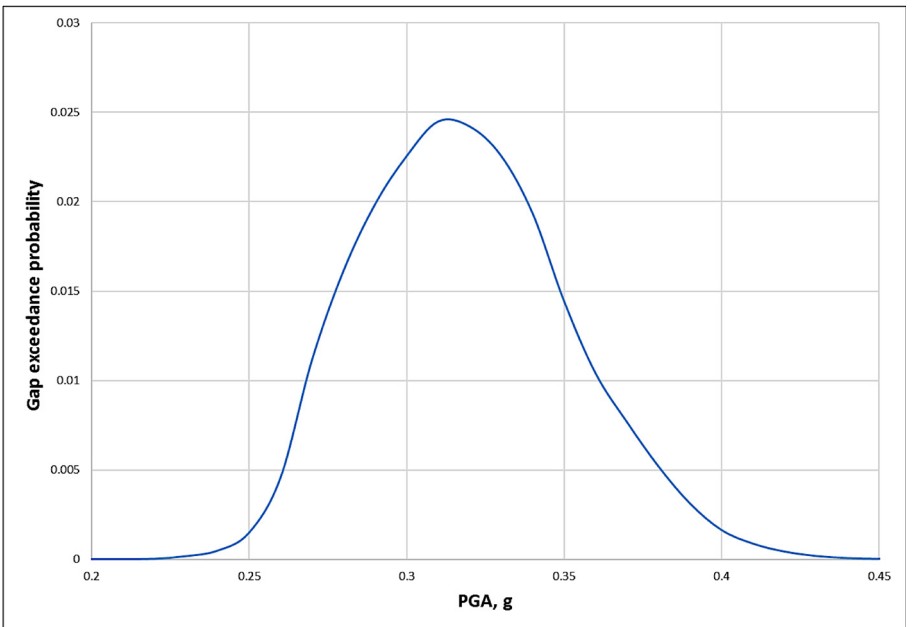

**Figure 10.** Exceedance probability distribution function of the gap versus peak ground acceleration.

Consequently, the loss of emergency service water system due to liquefaction can be integrated into the seismic PSA.

## 6. Discussion

In Section 3, two procedures were developed for integrating an earthquake-induced liquefaction hazard and its consequences into a seismic PSA.

These novel procedures for integration the liquefaction hazard and its consequences in the framework of a seismic probabilistic safety analysis were composed from results and experience gained from extensive analyses of the the Paks NPP and from published research results in the literature on liquefaction and liquefaction-induced settlement.

These procedures eliminate the methodological gaps that exist even in the basic industry guidance proposed by the EPRI [30] regarding the intensity parameter of the seismic hazard, the intensity parameter for liquefaction hazard, and the engineering demand parameter of the fragility for liquefaction.

The procedure presented in Section 3.1 is based on the hazard curves calculated via the probabilistic liquefaction hazard analysis proposed by Kramer and Mayfield [16] and on the evaluation of volumetric strain as proposed by Juang et al. [19]. This is a new procedure that allows the integration of liquefaction hazard into a seismic probabilistic safety analysis using the factor of safety to liquefaction as the intensity parameter for the hazard and the settlement as the engineering demand parameter for the fragility.

Despite the theoretical consistency of the new procedure, a more conventional methodology for the integration of liquefaction hazard into an SPSA has also been developed that uses peak ground acceleration as the intensity parameter of the seismic hazard and the engineering demand parameter as the fragility for liquefaction. This procedure is presented in Section 3.2. Here, certain conservative assumptions should be accepted for the magnitude scaling factor in the calculation of the factor of safety to liquefaction. The disaggregation of the seismic hazard into magnitude bins allows for the mean magnitude

of each hazard level to be defined, which can be used for the definition of the magnitude scaling factor. Thus, the peak ground acceleration remains as the only variable that defines the demand part in the liquefaction hazard evaluation. This procedure allows the integration of the liquefaction hazard into the seismic PSA in a conventional manner using the peak ground acceleration as the intensity parameter for the hazard and the engineering demand parameter for the fragility.

For the sake of conventions, in the case study for the Paks NPP, the formalism of expressing the settlement and tilt as a function of peak ground acceleration was used, as per Section 3.2. For failure criteria, the allowable settlement and tilt values were accepted for the main building. For the emergency service water piping that could be damaged by the difference between the settlement of the main building and the piping, closing the gap between the wall and the pipe at the wall penetration was assumed to be the failure criterion. This approach simplified the fragility evaluation. The uncertainty of the failure evaluation could be assessed, as described in Section 4. The results of the analysis could be compared and calibrated with results of the comprehensive soil-structure interaction analysis.

The proposed procedures can be used for evaluating the core damage frequency of the liquefaction-induced failure modes identified at the Paks NPP that have not been considered, up to now, by the nuclear industry guidelines and practice.

## 7. Conclusions

The procedures outlined in this study show new options for integrating liquefaction hazard and its consequent specific failure modes into a seismic PSA, for assuring safe operation of nuclear power plants at soil sites, where elimination of the liquefaction hazard, technically, is not practicable.

**Author Contributions:** Conceptualization, T.J.K. and Z.K.; methodology, T.J.K.; software, Z.K.; validation, Z.K.; formal analysis, Z.K.; investigation, T.J.K. and Z.K.; writing—original draft preparation, T.J.K.; writing—review and editing, T.J.K. and Z.K.; visualization, Z.K. All authors have read and agreed to the published version of the manuscript.

**Funding:** This research received no external funding.

**Institutional Review Board Statement:** Not applicable.

**Informed Consent Statement:** Not applicable.

**Data Availability Statement:** Data not presented in the paper are available at the authors.

**Conflicts of Interest:** The authors declare no conflict of interest.

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
