# Peer review of "Probabilistic Safety Analysis of the Liquefaction Hazard for a Nuclear Power Plant"

_geosciences, doi:10.3390/geosciences12050192_

Round 1

Reviewer 1 Report

The manuscript discussed the probabilistic analysis of liquefaction. It is interesting topic, but I seriously concern about the novelty of this research, which was not clearly presented in the manuscript. And there are other major issues:

  • There are some factors influencing liquefaction occurrence. The most common cause of liquefaction is the loss of shear strength of loose sandy soils under rapid load (i.e. undrained condition). However, this factor was not mentioned in this study. Therefore, I have a concern on the reliability of the probabilistic the authors proposed.
  • The PSA approach in this study is not new. The authors just adopted this approach and used the results from some case studies. In my opinion, this is not a significant contribution in the literature of liquefaction.
  • The authors also mentioned the settlement after liquefaction, but they did not have very good literature review on this issue. Perhaps, the authors focussed too much in the literature review on probabilistic analysis, but they missed to difficult the nature of liquefaction phenomenon. There have been increasingly number of investigations on this phenomenon, but the authors are not aware of them.
  • It was a bit confusing when the author performed the probabilistic analysis and showed the liquefaction hazard curves in Fig. 2. How can the authors come up with these curves? Are they related to the CPT-based method? Once again, there was not much insight discussion on the previous studies of liquefaction. And the curves in Fig. 2 were difficult to distinguish as well.

Reviewer 2 Report

This paper proposes a practical method for computing the annual probability of failure of critical structures due to liquefaction. Although the work is meaningful, it cannot be accepted without significant revisions. Therefore, it is strongly suggested to re-structure the manuscript and think carefully how to tell the story. The current manuscript involves many equations without showing their inter-relationships in terms of logic. In addition, the following comments may be useful for further improving the manuscript.

1- It is strongly suggested to re-structure the Abstract to highlight the background, main issue as well as the contribution and demonstration of this manuscript. In addition, it is suggested to use short sentences in a technical writing rather than long ones for reading convenience.

2- Page 1, Line 25: revise “…consequences that the vibration…” as “…consequences than the vibration…”

3- Which kind of nuclear plant is focused on, operating NPP or new ones? It is quite confusing in the current manuscript.

4-The authors may consider re-structure the introduction because the current one is lengthy and reviewer cannot differentiate readily the background information, literature review with the work by the authors.

5-Eq. (1): what is meant by a? Does it mean the horizontal acceleration rate? Better explanation regarding Eq.(1) is strongly needed.

6-Eq. (2): why a double-lognormal distribution is used for a? The authors stated that “conditional probability of failure” P(a). Which variable is conditional on when computing P(a)?

7-Because it is very difficult to get the framework and main idea of the manuscript, the current manuscript is suggested for publication with major and mandatory revisions.

Reviewer 3 Report

Manuscript titled "Probabilistic Safety Analysis for Liquefaction Hazard for a Nuclear Power Plant" written by Katona and Karsa proposes a probabilistic framework to assess the seismic performance of nuclear power plants against soil liquefaction. The study uses the work of Kramer and Mayfield as benchmark, which constitutes a sound starting point. Then, through the use of CPT-based settlement correlations, liquefaction-induced soil settlements are integrated to the framework by the Authors.

Overall, the work is useful and worth to be published following the modifications/amendments/explanations requested. Reviewer noted few errors and/or loss of fluency in the language (such as lines 134-136; lines 150-151; and similar), due to this reason, the manuscript needs at least one round of proofreading by a native speaker (or equivalent) who works in the field of geotechnical earthquake engineering.

Reviewer would like to separate the comments in terms of (A) major and (B) minor points.

(A) Major points
A.1. Section 2 describes the methodology, yet it is difficult to follow. It should be as generic and simple as possible. Instead of using case-specific input (e.g. Fig 2), the Authors are invited to generalize the contents (not only Fig 2 itself, but the entire section).
A.2. Section 2 should contain all the necessary input to be shown which is in this case: CPT-data, soil profile (CPT interpretation), FS-profiles, and similar). Then, the equations should be clearly stated in the order. A flowchart would be useful to assist the readers. In this flowchart, all possible implementations should be shown (such as CPT interpretation by Robertson, FS calculation by Idriss and Boulanger, computation of settlements by Zhang et al., Ishihara Yoshimine, Juang et al.). In the same flowchart, Authors are invited to highlight the part that is already proposed by Kramer and Mayfield to point the part of the methodology that is originally proposed by the Authors.
A.3. In the case study presented in Section 3, Authors discuss about tilting, however it is not explicitly stated in the previous section.
A.4.On the contrary to Section 2, Section 3 should be more specific to the case study, any generalization should be avoided. Furthermore, it is recommended that the transparency of the calculations should be increased by howing the calculations explicitly (not only the results), whenever possible.
A.5. This comment is rather general and not specific to any section. Users use liquefaction triggering approachs and corresponding settlement proxies that are suitable at free-field conditions. On the other hand, a bulky nuclear power plant may change these assumptions significantly. How would you think tht the analytical probabilistic approach would reflect the reality? Please elaborate this point within the text of the manuscript, as well.

(B) Minor points:
B.1. Line 45-46. Terminologies of incipent and gross are used through the manuscript. More frequently and globally used terms are cyclic mobility and flow liquefaction for those terms. Authors are invited to provide the correspondence in paranthesis in the first appearance, such as incipent (e.g. cyclic mobility) and gross (e.g. flow liquefaction).
B.2. Line 59, Line 60: Authors prefer event trees and fault trees. Those should be accompanied with event logic trees and fault logic trees. Please carefully check the entire manuscript.
B.3. Line 106, the parameters reflecting the methodology of Kramer are newly introduced and should be explained.
B.4. Paragraph between the lines 108 and 119 discusses a very interesting point. The Reviewer would like to kindly ask if a longer period spectral content (instead of PGA) would be more reasonably correlated to settlements?
B.5. Line 127: after EDP, there is a full stop ("."), maybe the intention was using comma (",") instead.
B.6. Conclusion seems very synthetic. Please extend.

Round 2

Reviewer 1 Report

I believe the authors addressed all comments from previous round.

Reviewer 2 Report

The authors have addressed most comments from the reviewer. Therefore, it is suggested for publication with the minor comments below:

What is meant by SSC? It should be defined first before using this as a shorthand in the manuscript.

Reviewer 3 Report

The Reviewer has read the revised version of the manuscript. The clarity and organization have been considerably improved. Although there is a good margin of improvement in terms of text fluency and figure quality , current version could be published. According to the Reviewer, the use of free field approaches to evaluate liquefaction in this case study poses a limitation, which could have been mentioned more strongly (but obviously it is up to the Authors' decision).